# Identification of Milk Adulteration in Camel Milk Using FT-Mid-Infrared Spectroscopy and Machine Learning Models

**DOI:** 10.3390/foods12244517

**Published:** 2023-12-18

**Authors:** Zhiqiu Yao, Xinxin Zhang, Pei Nie, Haimiao Lv, Ying Yang, Wenna Zou, Liguo Yang

**Affiliations:** 1National Center for International Research on Animal Genetics, Breeding and Reproduction (NCIRAGBR), Ministry of Science and Technology of the People’s Republic of China, Huazhong Agricultural University, Wuhan 430070, China; 2Key Laboratory of Animal Genetics, Breeding and Reproduction, Ministry of Education, College of Animal Science and Technology, Huazhong Agricultural University, Wuhan 430070, China; 3College of Veterinary Medicine, Hunan Agricultural University, Changsha 410128, China

**Keywords:** camel milk, adulteration, Linear Discriminant Analysis, neural network, FT-MIR, chemometric analysis

## Abstract

Camel milk, esteemed for its high nutritional value, has long been a subject of interest. However, the adulteration of camel milk with cow milk poses a significant threat to food quality and safety. Fourier-transform infrared spectroscopy (FT-MIR) has emerged as a rapid method for the detection and quantification of cow milk adulteration. Nevertheless, its effectiveness in conveniently detecting adulteration in camel milk remains to be determined. Camel milk samples were collected from Alxa League, Inner Mongolia, China, and were supplemented with varying concentrations of cow milk samples. Spectra were acquired using the FOSS FT6000 spectrometer, and a diverse set of machine learning models was employed to detect cow milk adulteration in camel milk. Our results demonstrate that the Linear Discriminant Analysis (LDA) model effectively distinguishes pure camel milk from adulterated samples, maintaining a 100% detection rate even at cow milk addition levels of 10 g/100 g. The neural network quantitative model for cow milk adulteration in camel milk exhibited a detection limit of 3.27 g/100 g and a quantification limit of 10.90 g/100 g. The quantitative model demonstrated excellent precision and accuracy within the range of 10–90 g/100 g of adulteration. This study highlights the potential of FT-MIR spectroscopy in conjunction with machine learning techniques for ensuring the authenticity and quality of camel milk, thus addressing concerns related to food integrity and consumer safety.

## 1. Introduction

With the improvement of living standards, camel milk has garnered escalating attention and favor due to its exceptional nutritional profile. Camel milk is enriched with essential amino acids, such as proteins, glutamic acid, and lysine, which are vital components of the human diet. Moreover, it boasts a 5% lower calorie content compared to conventional cow’s milk [1]. Camel milk has similar calcium and iron content to cow’s milk, but higher levels of sodium, vitamin C, and niacin [2,3]. Compared to water buffalo milk, camel milk contains a higher proportion of αS1-casein, αS2-casein, and κ-casein in its casein content, potentially offering consumers improved digestion and anti-allergic properties [4]. Research indicates its potential in disease resistance, including alleviating liver ailments, treating gastric ulcers, and enhancing female ovarian ovulation [5,6]. Despite its high nutritional value, camel milk remains scarce in the market, commanding a premium price, often dubbed as “desert gold”. Unfortunately, unscrupulous vendors have begun adulterating camel milk with cheaper commercial milks, including goat, cow, buffalo, and others [7,8]. The remarkably similar texture and composition between camel milk and other milks make conventional methods inadequate for detecting adulteration effectively.

Fourier Transform Mid-Infrared Spectroscopy (FT-MIR) plays a pivotal role in the detection of adulteration in dairy products. Infrared spectroscopy analysis is an invaluable technique for determining molecular components in samples by assessing their absorption, scattering, or transmission of infrared light. To achieve this, samples are exposed to an infrared light source, and their absorption of infrared light at different wavelengths is measured. These absorption peaks correspond to the vibrational and stretching modes of different molecules within the sample, each possessing a unique infrared spectroscopic fingerprint [9]. Consequently, by analyzing the infrared spectra of samples, their composition and quality can be ascertained, enabling the detection of illicit adulteration. FT-MIR spectroscopy is instrumental in identifying adulterants such as melamine, plant fats, sucrose, and formaldehyde in dairy products [10,11,12]. Thanks to the characteristics of FT-MIR analysis, including speed, non-destructiveness, and high sensitivity, it allows for the rapid testing of large sample batches within a short time frame. This proves particularly advantageous for real-time quality control during the production process and the swift detection of adulteration issues. Consequently, establishing an efficient method to detect cow milk adulteration in camel milk is of paramount importance.

In the realm of infrared spectrum classification and quantitative models, prediction accuracy can vary significantly based on the chosen preprocessing methods and algorithms applied to the spectra. Currently, widely adopted models for predicting milk composition and detecting adulteration include Partial Least Squares (PLS) and Principal Component Analysis (PCA) [13,14,15,16]. Madhusudan et al. successfully constructed a near-infrared spectroscopy adulteration detection model for turmeric using PLS, random forests, and decision tree regression among other machine learning algorithms. The PCA algorithm was utilized for spectral data dimensionality reduction in their study [17]. As computational power and machine learning methods have advanced, multivariate models are increasingly employed for calibrating component concentrations in milk. Recent studies have explored the use of Support Vector machines (SVM) and Artificial Neural Networks (ANN) in FT-MIR spectroscopic analysis [18,19]. Ana et al. employed a convolutional neural network-based image method to identify water adulteration in milk, achieving an accuracy of 93% [20]. Nadia et al. successfully established a sesame oil adulteration model using an electronic nose combined with SVM and ANN, where the SVM model exhibited slightly superior sensitivity and specificity compared to the ANN model [21]. These relatively new technologies offer advantages such as excellent generalization capabilities, the ability to model non-linear data, high scalability, and ease of training [22].

The primary objective of this study is to identify milk adulteration in camel milk using FT-MIR spectroscopy. To achieve this objective, camel milk from Alxa in Inner Mongolia was deliberately adulterated with a concentration gradient of cow milk ranging from 10–90 g/100 g. Subsequently, the FOSS FT-6000 FT-MIR milk component analyzer was employed, utilizing PCA and Linear Discriminant Analysis (LDA) as analytical tools to distinguish adulterated samples from pure camel milk. Additionally, we utilized four machine learning methods: PLS, PCA, SVM, and ANN to build models that could quantify the extent of milk adulteration in camel milk. This comprehensive approach leverages both traditional statistical methods and cutting-edge machine learning techniques to enhance the accuracy and reliability of detecting milk adulteration in camel milk using FT-MIR spectroscopy.

## 2. Materials and Methods

### 2.1. Sample Collection

Twenty-four camel milk samples were collected from Alxa League, Inner Mongolia in China, between July and August 2023. Additionally, 50 Holstein cow milk samples were obtained from Wuhan, Hubei. All milk samples were preserved at −20 °C after the addition of 0.1% potassium dichromate. The camel milk samples were randomly adulterated with cow milk at concentrations of 10 g/100 g, 20 g/100 g, 50 g/100 g, 70 g/100 g, and 90 g/100 g after mixing. Each adulterated sample concentration was prepared in seven replicates, with each sample containing 30 mL. In total, there were 35 adulterated camel milk samples, 34 cow milk samples, and 24 pure camel milk samples, totaling 93 samples for subsequent FT-MIR analysis.

### 2.2. FT-MIR Analysis and Data Preprocessing

After rapid thawing at 40 °C in a water bath, milk samples were analyzed using the MilkoScan FT-6000 instrument (FOSS Analytical A/S, Hillerød, Denmark) to obtain the FT-MIR spectra for each sample. Each sample is scanned twice and the results are averaged to analyze fat, protein, lactose and other milk components. The FT-MIR spectra incorporated a region extending from 926 to 5012 cm^−1^. However, the O-H bending region (1600–1710 cm^−1^) and the O-H stretching region (3020–5012 cm^−1^) were omitted due to the disturbances induced by the water content in milk. The remnant spectra region underwent further analysis. For the refinement of the raw spectra, a combination of seven distinct preprocessing techniques was employed encompassing Standard Normal Variate (SNV), the 11-point Savitzky–Golay (SG) method, First Derivative plus Savitzky–Golay (SG1), Second Derivative plus Savitzky–Golay (SG2), SNV plus Savitzky–Golay (SSG), SNV plus Savitzky–Golay plus First Derivative (SSG1), and SNV plus Savitzky–Golay algorithm plus Second Derivative (SSG2) (Figure 1). These preprocessing steps were facilitated by the employment of R packages “prospect” (version 0.26) and “baseline” (version 1.3-4).

### 2.3. Data Preprocessing and Model Building

The collected FT-MIR spectral data were standardized to ensure that the spectral intensities at each wavenumber had zero mean and unit variance. The standardization is performed using the SCALE function in the R programming language, and the formula is as follows
(1)$$Z=\frac{\left(X-\mu \right)}{\sigma }$$

*Z* is the standardized value, *X* is the original data point value, *μ* is the mean of the dataset, *σ* is the standard deviation of the dataset.

The dataset was randomly split into a training set (70%) for model construction and a test set (30%) for performance evaluation. PCA, an unsupervised learning method, was employed to reduce the dimensionality of the data while retaining maximum variance. LDA, being a supervised learning method, aims to optimize the separation between sample classes while simultaneously reducing data dimensionality. The PCA and LDA models were constructed using the training set, and their performance on the test set was evaluated, including metrics like classification accuracy, sensitivity, and specificity. To quantify the adulteration level of cow milk in camel milk, quantitative models were constructed using machine learning. Two linear regression models, Partial Least Squares Regression (PLSR) and PCA, as well as two non-linear models, Support Vector Machine (SVM) and Artificial Neural Network (ANN), were utilized. PLSR is a chemometric method widely used in spectral data analysis, decomposing spectral data into Latent Variables (LV) to explain observed variances. SVM, a method for binary classification in supervised learning environments, is effective even when dealing with high-dimensional features, where the feature count exceeds the sample count [23,24]. ANN, representing a non-linear extension of traditional linear regression models, can model complex non-linear relationships by utilizing hidden layers [25]. Parameters for each model were optimized using Cross Validation (CV) statistics and the “expand.grid” function.

The performance of each model was evaluated using internal 10-fold cross-validation statistics, including Root Mean Square Error (RMSE) and coefficients of determination (R^2^). The parameter “selectionFunction = ‘best’” was used to select optimized values for factors in the “train” function of the CARET package, indicating the model with the lowest cross-validation RMSE (RMSEcv) was chosen. For PLSR and PCR, the maximum number of latent variables was set to 25. SVM’s C-test values ranged from 0.0001 to 2.5. ANN’s hidden unit count varied from 1 to 5, and decay values tested were 0, 0.0001, 0.001, 0.01, 0.1, 0.2, 0.3, 0.4, and 0.5. In the realm of the CARET package, the “size” parameter denotes the quantity of units within a hidden layer, while the “decay” parameter signifies the degree of regularization strength. For both PLSR and PCR, the upper limit for the latent variables was established at 25. The evaluation of model performance encompassed metrics such as Root Mean Squared Error of Cross-Validation (RMSECV) and the coefficient of determination (R2cv), determined through internal 10-fold cross-validation statistics. Model validation was conducted by estimating the Root Mean Squared Prediction Error (RMSEP) on an external test set. All machine learning algorithms were implemented in R using the CARET package version 6.0-93 (version 4.2.2; https://www.r-project.org/ (accessed on 10 September 2023)) [26].

### 2.4. Quality Control for the Method

The devised approach was subjected to validation following the guidelines outlined in the International Conference on Harmonization (ICH) Q2 (R1). The Limit of Detection (LOD) was determined through analysis of six pure camel milk samples, with the standard deviation of the matrix being calculated for this purpose [27].

In terms of relative bias, recovery and repeatability, the validation protocol referenced our previous study [28]. Five different levels of milk adulteration in camel milk (10 g/100 g, 20 g/100 g, 50 g/100 g, 70 g/100 g, and 90 g/100 g) were studied, with each level replicated three times, resulting in a total of 15 samples. The formulas for “*Bias* (%)”, “*Recovery* (%)”, and “*Repeatability* (*RSD*%)” are as follows:(2)$$Bias\left(\%\right)=\frac{\mathrm{Measured}\;\mathrm{Value}-\mathrm{Reference}\;\mathrm{Value}}{\mathrm{Reference}\;\mathrm{Value}}\times 100$$
(3)$$Recovery\left(\%\right)=\frac{\mathrm{Measured}\;\mathrm{Value}}{\mathrm{Reference}\;\mathrm{Value}}\times 100$$
(4)$$Recovery\left(\%\right)=\frac{\mathrm{Standard}\;\mathrm{Deviation}}{\mathrm{Mean}\;\mathrm{Value}}\times 100$$

## 3. Results and Discussion

### 3.1. FT-MIR of Camel Milk, Cow’s Milk and Adulterated Milk

The spectra obtained using FT-6000, as shown in Figure 2, exhibit noticeable noise patterns in two regions: 1600–1700 cm^−1^ and 3020–3400 cm^−1^, which are associated with O-H vibrations from H_2_O. Consequently, in studies employing mid-infrared spectroscopy, the water absorption regions are typically excluded to enhance measurement accuracy. All spectra curves of camel milk samples (red) closely resemble those of cow milk (blue). Particularly, when camel milk is mixed with cow milk (green), the chemical composition of the two milk matrices is highly similar. Minor differences in the mid-infrared spectra are observed in the 1076 and 1550 cm^−1^ regions. The primary components of camel milk can be correlated with characteristic spectral bands in the FT-MIR spectra of cow milk, as their chemical compositions and bonds absorb spectra at specific wavenumbers. Regions related to lactose include C-O, C-C, and C-H stretching vibrations around 1076 cm^−1^, as well as C-O-C stretching vibrations at 1157 and 1250 cm^−1^ [29]. Protein-related regions appear near 1550 cm^−1^ with C-N and N-N stretching peaks. Regions associated with fatty chains are observed around 1390 and 1454 cm^−1^ with C-H twisting vibrations of -CH3 and -CH2, and C-H stretching of -CH3 and -CH2 at 2862 and 2927 cm^−1^. Additionally, a region related to fats is present near 1743 cm^−1^ due to C=O bond stretching [30]. Hence, the disparities observed around 1076 and 1550 cm^−1^ are attributed to differences in lactose and lipid content between camel and cow milk.

The results of milk component analysis using FT-MIR are summarized in Table 1.

It is evident that Camel Milk exhibits higher milk fat and milk protein content compared to Cow Milk. Camel Milk has a milk fat content of 3.91 ± 0.46% and a milk protein content of 3.69 ± 0.20%, while Cow Milk has a milk fat content of 3.40 ± 0.33% and a milk protein content of 3.19 ± 0.23%. The milk fat and milk protein in Camel Milk are approximately 15.1% and 15.5% higher than those in Cow Milk. Previous studies have reported that the fat content in Camel Milk ranges from 1.2% to 6.4%, with an average of 3.5 ± 1.0% [31]. The relatively higher milk fat content in the Camel Milk samples we collected may be attributed to various factors, including environmental factors such as analytical measurement procedures, geographical location, feeding conditions, and breeds, as well as physiological factors such as lactation stage, age and parity [32,33]. Geographic origin and seasonal variations are among the primary factors influencing Camel Milk composition [34]. For instance, research indicates that the fat, protein, and solids content in Asian camel milk is significantly higher compared to North Africa, India, and West Asia [31], Additionally, the protein content in camel milk during the spring season surpasses that of other seasons [35].

A strong positive correlation exists between fat and protein content in Camel Milk [36]. The total protein content in Camel Milk ranges from approximately 2.15% to 4.90%, with an average of 3.1 ± 0.5%. As discussed earlier regarding milk fat content, factors influencing milk protein content in Camel Milk exhibit similar variations. Casein is the predominant protein in Camel Milk, constituting approximately 1.63–2.76% of the total protein content, which corresponds to 52–87% of the total protein [37]. β-casein is the major camel milk casein protein, followed by αs1-casein, constituting 65% and 21% of the total whey proteins, respectively [4]. Previous studies have indicated that the lactose content in camel milk is significantly higher during the rainy season, approximately 5.57 ± 0.15%, and lower during the winter season, around 4.58 ± 0.09%. Additionally, the urea content in camel milk typically falls within the range of 33–45 mg/100 g [38]. Camel milk is known to have lower levels of saturated fatty acids compared to cow’s milk, with higher concentrations of long-chain fatty acids [39]. Our research results are consistent with the results described above.

### 3.2. Identification of Adulteration in Camel Milk with Cow Milk

FT-MIR spectroscopy was employed as the detection technique to analyze milk adulteration, differentiating between unadulterated camel milk, cow’s milk, and adulterated camel milk samples. Initially, exploratory analysis using PCA was conducted to assess the inherent variability in the samples. Following an investigation of all eight preprocessing methods, PCA failed to effectively distinguish adulterated camel milk samples, with only the SG2 preprocessing method showing some separation between camel milk and cow’s milk (Figure 3a). For the spectra data processed with SG2, Principal Component 1 and Principal Component 2 explained variances of 33.2% and 24.3%, respectively, with the first five principal components collectively explaining 74.2% of the data variance.

To address this issue, LDA, a supervised pattern recognition technique, was applied to the FT-MIR spectral data. We compared the effectiveness of various preprocessing methods, as shown in Figure 3b. LDA, after SG preprocessing, successfully identified pure camel milk, pure cow milk, and camel milk adulterated with cow milk, achieving 100% accuracy, sensitivity, and specificity. Even at the lowest level of adulteration (10%), clear identification was achieved. The substantial difference in performance between the two models can be attributed to the nature of LDA as a supervised learning method. Its primary objective is to find projection directions that maximize the differences between different categories, thereby preserving maximum category information during dimensionality reduction [40]. In contrast, PCA is an unsupervised learning method primarily focused on finding directions with the maximum variance in the data, regardless of category information. Consequently, PCA primarily considers the overall data structure during dimensionality reduction and may not be well-suited for detecting adulteration as adulterated samples may exhibit similar variances in data space compared to genuine samples [41].

### 3.3. Determination of Adulteration in Camel Milk with Cow Milk

The reference value range for determining the cow milk content in adulterated camel milk is 0 to 100 g/100 g. This reference range is then used for constructing and predicting models using different FT-MIR spectra with PLS, PCA, SVM, and ANN. The results obtained for the quantification of the adulteration levels are presented in Table 2.

In Table 2, a clear pattern emerges, highlighting that the majority of optimal predictive models favor the use of ANN. This inclination can be primarily attributed to the incorporation of ANN’s hidden layers, which, in contrast to conventional statistical regression models, take into account the intricate non-linear interactions present within distinct spectral regions. The concept of statistical interaction goes beyond the mere additive effects of multiple input variables on an output variable, revealing the collective impact of these variables [42]. Within the realm of ANN, these statistical interplays among input variables unfold within its hidden layers, facilitated by the transmission of weights from inputs to these concealed nodes [43]. Furthermore, the adoption of logic functions as activation mechanisms enhances the interconnected non-linearity and interaction among input variables within the ANN model. Therefore, the complex web of interactions and non-linearity inherently resides within the ANN model, providing it with a theoretical advantage over linear regression models like PLS. Once the input–output variables are defined, the architecture of the neural network heavily relies on the number of neurons in its hidden layers. It is crucial to exercise restraint when increasing neurons in ANN, as an excessive proliferation can lead to overtraining, ultimately compromising the network’s ability to generalize and resulting in overfitting. Most of the fatty acid prediction models constructed using ANN showed minimal RMSE when comprising one–three hidden layers. Moreover, the judicious application of modest weight decay helps mitigate overfitting tendencies, as evidenced by the convergence of RMSE values between the training and testing datasets.

According to the ICH Q2 (R1) guidelines, the method for detecting cow milk adulteration in camel milk using FT-MIR spectroscopy establishes the Limit of Detection (LOD) and Limit of Quantification (LOQ) based on the standard deviation observed in pure camel milk samples (*n* = 6). The calculated LOD is 3.27 g/100 g, and the LOQ is 10.90 g/100 g.

Furthermore, the FT-MIR method was rigorously validated to assess its authenticity and precision in detecting cow milk adulteration within camel milk, adhering to the guidelines outlined in ICH Q2 (R1). Table 3 presents compelling evidence of the method’s robustness, showcasing minimal relative bias and remarkable recovery rates across a concentration range spanning from 10–100 g/100 g. The average bias remained within the range of 0.15% to 7.19%, while the recovery rates were consistently between 97.02% and 107.19%. The relative standard deviation (RSD) exhibited values spanning from 0.42% to 6.79%. Repeatability for adulterated cow milk concentrations of 20 g/100 g, 50 g/100 g, and 70 g/100 g was well below 5%. However, at an adulteration level of 10 g/100 g, the Relative Standard Deviation (RSD) surpasses 5%, indicating potential escalation in both systematic and random errors within the prediction model at lower concentrations [44]. The correlation between the predicted concentrations and the true concentrations was assessed using the linear equation: y = 1.0004x + 0.5534, yielding an R^2^ value of 0.9979 (Figure 4a). The slope and R^2^ values of the linear equation illustrate the robust agreement between the MIR predictions and the reference values.

Figure 4b displays the Bland–Altman scatter plot, evaluating the predicted values from the model against the true values. The y-axis represents the differences between the values obtained by the two methods, while the x-axis represents the average of these measured values. The Bland–Altman plot indicates that the adulteration values predicted by the FT-MIR model are slightly positive, with a mean difference of 0.5514. The 95% limits of agreement range from −3.18 to 4.28 g/100 g. At both the adulteration concentration of 90 g/100 g and in pure milk, our adulteration model predicted an adulteration quantity for one sample that exceeded the 95% confidence interval. The occurrence of this phenomenon may stem from various factors. Specifically addressing the ANN model, renowned for its capacity to learn and adapt to non-linear relationships. However, their adaptability encounters challenges in extreme cases involving sparse data. The observed trend of the relative deviation initially decreasing and subsequently increasing with the adulteration concentration signifies this challenge. As the adulteration concentration increases from low to high, the neural network is likely more adept at capturing non-linear relationships between different features, thereby reducing relative deviation. Generally, the predictive performance of the model tends to improve with an increase in data volume, including the incorporation of more extreme case data. In our future work, we plan to explore strategies such as increasing the sample size, adjusting hyperparameters, or adopting alternative spectral preprocessing techniques to enhance the consistency of the approach.

Table 4 summarizes the performance comparison of different quantitative methods for detecting adulteration of cow milk in camel milk.

## 4. Conclusions

In this study, we demonstrated the effectiveness of FT-MIR spectroscopy in identifying and quantifying adulteration in camel milk with cow milk. By using pre-processed FT-MIR spectra and LDA, we obtained optimal classification results. Furthermore, our Artificial Neural Network (ANN) models showed excellent quantitative accuracy for adulterated cow milk. These models achieved high levels of accuracy and precision, establishing the proposed method as a valuable, rapid, and non-destructive tool for screening camel milk quality.

Future research can enhance the methodology’s applicability by expanding the dataset to include diverse geographical regions and varying levels of adulteration. Exploring alternative spectroscopic techniques or combining multiple analytical methods may offer a more comprehensive understanding of milk adulteration. Additionally, refining model parameters, such as adjusting hyperparameters in ANN or exploring different pre-processing strategies, could further optimize the methodology’s performance, ensuring its continued effectiveness in dairy quality control.

## Figures and Tables

**Figure 1 foods-12-04517-f001:**
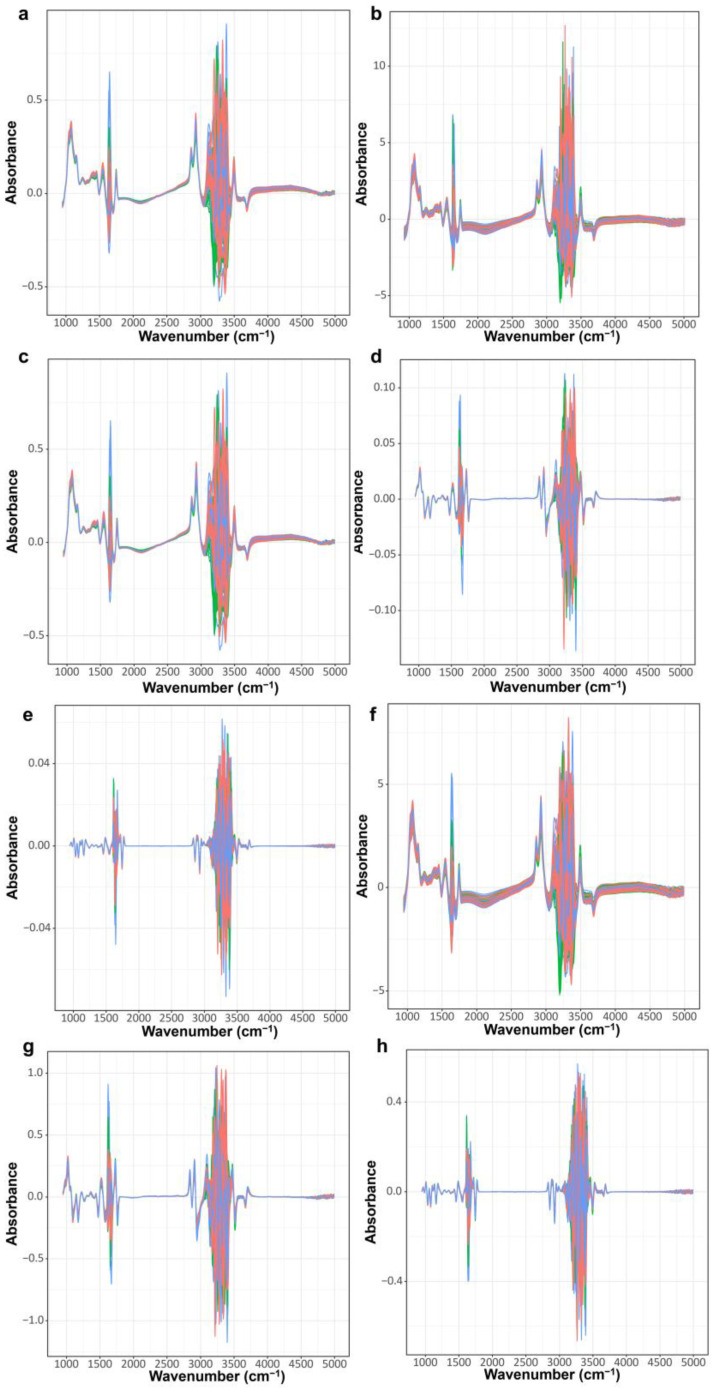
Mid-infrared spectra after various preprocessing methods. (**a**) Raw spectra (Base); (**b**) Mid-infrared spectra after standard normal variables processing (SNV); (**c**) Mid-infrared spectra after Savitzky–Golay algorithm processing (SG); (**d**) Mid-infrared spectra after first derivative and Savitzky–Golay algorithm processing (SG1); (**e**) Mid-infrared spectra after second derivative and Savitzky–Golay algorithm processing (SG2); (**f**) Mid-infrared spectra after SNV and Savitzky–Golay algorithm processing (SSG); (**g**) Mid-infrared spectra after SNV, Savitzky–Golay algorithm, and first derivative processing (SSG1); (**h**) Mid-infrared spectra after SNV, second derivative, and Savitzky–Golay algorithm processing (SSG2).

**Figure 2 foods-12-04517-f002:**
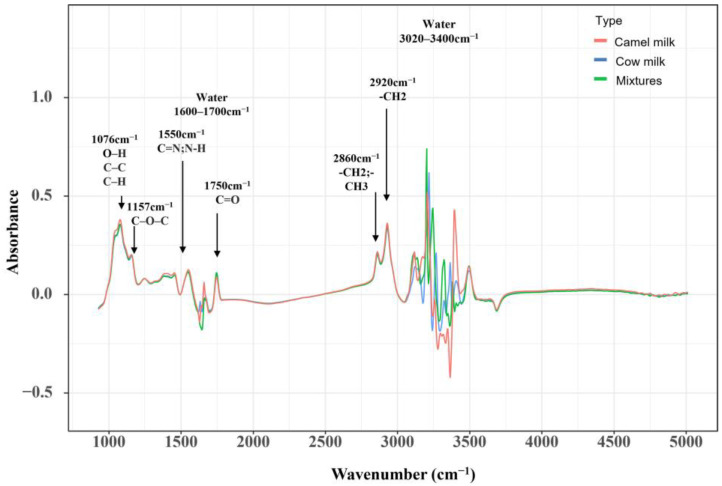
FT-MIR Spectra of Camel Milk (Red), Cow Milk (Blue), and Their Mixtures (Green).

**Figure 3 foods-12-04517-f003:**
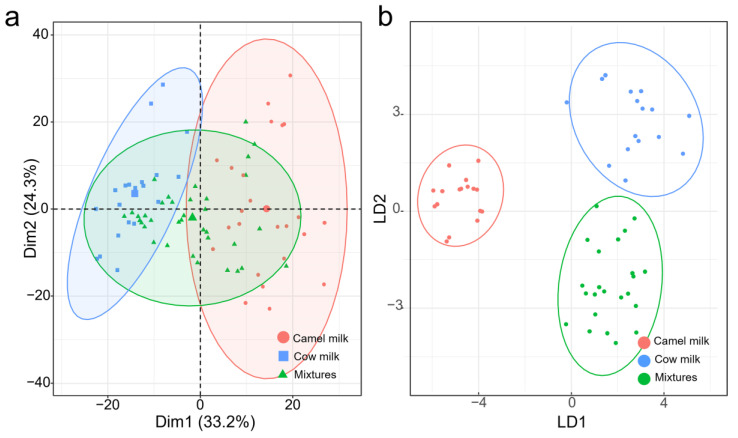
PCA score and LDA response plot of FT-MIR spectra after preprocessing. (**a**) PCA score plot of FT-MIR spectra after SG2 preprocessing. (**b**) LDA response plot of FT-MIR spectra after SG preprocessing.

**Figure 4 foods-12-04517-f004:**
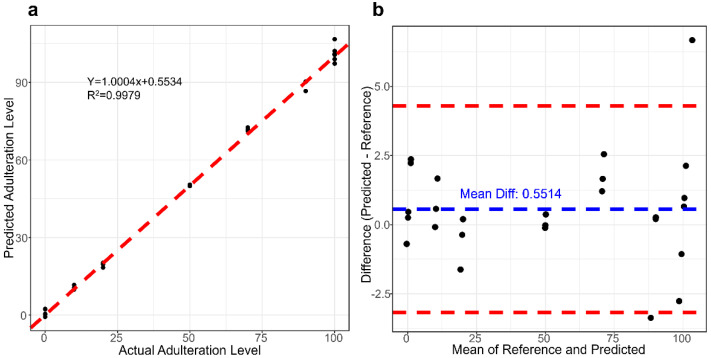
Comparison of Predicted Milk Adulteration Levels in Camel Milk using the SG-ANN Model against the Reference Values. (**a**) Regression curve of reference values and predictions. (**b**) Bland–Altman plot of reference values and predictions. The blue dashed line represents the mean difference, and the red dashed lines represent the 95% confidence interval.

**Table 1 foods-12-04517-t001:** Descriptive Statistics for Camel Milk and Cow Milk.

Item	Camel Milk (Mean ± SD)	Coefficient of Variation %	Cow Milk (Mean ± SD)	Coefficient of Variation %	Adulterated Milk (Mean ± SD)	Coefficient of Variation %
Fat (%)	3.91 ± 0.46	11.89	3.40 ± 0.33	9.90	3.58 ± 0.31	8.85
Protein (%)	3.69 ± 0.20	5.43	3.19 ± 0.23	7.34	3.41 ± 0.21	6.02
Lactose (%)	5.63 ± 0.27	4.79	5.16 ± 0.16	3.13	5.37 ± 0.20	3.79
TS (%)	14.15 ± 0.53	3.75	12.46 ± 0.53	4.26	13.19 ± 4.68	4.68
SNF (%)	10.07 ± 0.36	3.61	8.92 ± 0.34	3.91	9.43 ± 0.41	4.38
Urea (mg/100 mL)	36.58 ± 8.62	23.57	13.08 ± 1.26	9.67	24.57 ± 8.56	34.84
β-Casein (%)	2.89 ± 0.14	4.83	2.54 ± 0.19	7.79	2.66 ± 0.13	5.08
SFA (g/100 g)	2.23 ± 0.43	17.78	2.25 ± 0.26	11.51	2.29 ± 0.25	11.24
MUFA (g/100 g)	1.21 ± 0.39	32.72	0.85 ± 0.13	15.8	1.02 ± 0.22	22.33
PUFA (g/100 g)	0.09 ± 0.03	33.32	0.08 ± 0.01	20.58	0.07 ± 0.03	73.57

**Table 2 foods-12-04517-t002:** Results of cow milk determination in camel milk using various pre-processed ft-mir data and machine learning.

Re-Processing	Model Type	LV ^1^	RMSE_CV_ ^2^	R^2^_cv_ ^3^	RMSE_P_ ^4^	R^2^_P_ ^5^	PRD ^6^
Base	PLS	19	4.456	0.991	3.404	0.993	12.380
	PCA	25	5.366	0.987	3.528	0.992	11.947
	SVM	0.5	6.188	0.983	3.289	0.993	12.814
	**ANN**	**3; 0.01**	**4.318**	**0.992**	**3.298**	**0.994**	**12.778**
SG	PLS	14	4.765	0.989	3.181	0.994	13.243
	PCA	25	5.320	0.986	3.433	0.993	12.276
	SVM	0.5	6.162	0.983	3.406	0.993	12.374
	**ANN**	**3; 0.01**	**4.302**	**0.993**	**2.902**	**0.995**	**14.526**
SG1	PLS	14	5.525	0.986	3.818	0.991	11.039
	PCA	25	5.998	0.984	3.926	0.991	10.374
	SVM	0.05	7.426	0.982	4.687	0.981	8.992
	**ANN**	**6; 0.001**	**4.695**	**0.992**	**3.634**	**0.992**	**11.413**
SG2	PLS	8	6.997	0.980	4.417	0.980	9.543
	PCA	16	6.879	0.982	4.587	0.988	9.188
	SVM	0.001	6.747	0.986	6.493	0.976	6.493
	**ANN**	**6; 0.001**	**5.597**	**0.988**	**3.747**	**0.992**	**11.249**
SNV	PLS	14	7.793	0.959	5.459	0.983	7.721
	PCA	25	7.979	0.966	5.570	0.982	7.567
	SVM	2.5	7.841	0.966	6.632	0.975	6.354
	**ANN**	**3; 0.3**	**6.872**	**0.976**	**3.410**	**0.993**	**12.359**
SSG	PLS	13	7.840	0.958	5.553	0.982	7.591
	PCA	25	8.076	0.963	5.553	0.982	7.591
	SVM	2.5	7.820	0.967	6.795	0.974	6.203
	**ANN**	**5; 0.4**	**5.479**	**0.987**	**3.083**	**0.994**	**13.672**
SSG1	PLS	25	31.281	0.655	33.286	0.376	1.266
	PCA	18	35.301	0.389	36.297	0.259	1.611
	**SVM**	**0.005**	**11.774**	**0.949**	**9.594**	**0.948**	**4.393**
	ANN	3; 0.5	27.937	0.576	36.101	0.266	1.167
SSG2	PLS	2	33.484	0.433	39.983	0.100	1.054
	PCA	8	33.689	0.422	38.610	0.161	1.091
	**SVM**	**0.005**	**11.190**	**0.962**	**9.954**	**0.944**	**4.234**
	ANN	3; 0.5	31.159	0.511	34.193	0.342	1.122

Note: ^1^ LV: Latent variables used in the model. ^2^ RMSECV: Root mean square error of cross-validation; ^3^ R^2^_CV_: Coefficient of correlation for cross-validation. ^4^ RMSE_P_: Root mean square error of prediction. ^5^ RP: Coefficient of correlation for prediction. ^6^ RPD: Ratio performance to deviation for prediction.

**Table 3 foods-12-04517-t003:** Trueness and precision results for each concentration level in the validation data.

Theoretical Value	Predicted Value	Trueness		Precision
Level (g/100 g)	Mean ± SD (%)	Bias (%)	Recovery (%)	Repeatability (RSD%)
10	10.71 ± 0.72	7.19	107.19	6.75
20	19.40 ± 0.76	2.97	97.02	3.91
50	50.07 ± 0.21	0.15	100.15	0.42
70	71.80 ± 0.55	2.57	102.57	0.77
90	89.03 ± 1.69	1.07	98.92	1.90

**Table 4 foods-12-04517-t004:** Detection method for cow milk adulteration in camel milk.

Technique	Advantages	Disadvantages	Detection Effect	References
PCR	Very selective and sensitive	The sample DNA extraction stage requires contamination prevention, and specific primers need to be designed and synthesized.	Recoveries ranging from 80% to 110% with a coefficient of variation of less than 7%	Wu et al. [44]
Ultra-high performance liquid chromatography	High Resolution High Sensitivity	Expensive Equipment Complex Sample Pre-treatment	Recoveries ranging from 94% to 105% with a coefficient of variation of less than 5%	Li et al. [7]
NIR spectroscopy	Convenient, rapid, automated and simplify sample handling	Limited sensitivity Expensive instrumentation.	The detection limit is 0.5%, and the quantification limit is 2%. The R-squared value is 0.94.	Mabood et al. [15]
FTIR spectroscopy	Convenient, rapid, automated and simplify sample handling	Limited sensitivity Expensive instrumentation.	The relative error is 3.8%, and the detection limit is 2.59%. The R-squared value is 0.994.	Souhassou et al. [45]
Electrochemical sensor	Good speed, sensitivity and stability	Expensive instrumentation.	Identification of β-lactoglobulin within the range of 4–100 ng/mL, with a detection limit of 3.58 ng/mL.	Meng et al. [46]

## Data Availability

Data is contained within the article.

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
