# Peer review of "Identification of Milk Adulteration in Camel Milk Using FT-Mid-Infrared Spectroscopy and Machine Learning Models"

_foods, 2023, doi:10.3390/foods12244517_

Round 1
Reviewer 1 Report
Comments and Suggestions for Authors
The authors are not very up-to-dated regarding camel milk composition and state so many mistakes and un correct references. In the order:
Line 32: camel milk composition in minerals or vitamins must be described based on critical review because camel milk is poor in vitamin A and most vitamins B, and not so rich in certain minerals (see the reviews of Konuspayeva et al., 2022. Mineral status in camel milk: a critical review. Animal Frontiers, 12(4), 52-60.)
Line 33/ PrP is specific protein but its antibacterial role is unknown.
Line 37: It is not true that camel milk is without beta-casein!! (see Kappeler et al., 1998. Sequence analysis of Camelus dromedarius milk caseins. J. Dairy Res., 65(2), 209-222). At reverse, it contains higher proportion than in cow milk.... And despite the lack of Beta-lactoglobulin, there is some case of camel milk allergy (see Ehlayel and Bener, 2018. Camel milk allergy. Allerg. Asthma Proc., 39(5), 384-388)
Lines 178-189. The values of fat and proteins are highly variable according to the regions in the world (See your reference 28 and a more comprehensive data from the same author: Konuspayeva, 2020, Camel milk composition and nutritional value. In "Handbook on health and environmental benefits of camel products. IGI Global, Hershey, USA, 15-40). Regarding the physiological variability, your reference is not convenient because the tables in this paper did not give the values according to parity or physiological status. Better to cite Musaad et al., 2013. Seasonal and physiological variation of gross composition of camel milk in Saudi Arabia. Emir. J. Food Agric., 25(8), 618-624
Line 196 alphaS1-casein is a casein, not a whey protein, even if it is possible to find a part of this casein in the lactoserum.
But the most important point is regarding the interest of the method by using sophisticated equipment (FT-mid IR spectroscopy) to detect alduterated milk. Indeed, there is already easy commercial kit to detect other milk in camel milk, by testing the presence of Beta-lactoglobulin normally absent in camel milk. The method used by this commercial kit is based on the Radial Immunodiffusion test (RID test) detecting the presence of Beta-lactoglobulin by a reaction antigen-antibody provoking a precipitation of the protein if it is present in the milk.
Specific comments
Line 82: never start a sentence with numbering; Write Twenty-four camels
Author Response
Thank you very much for your thorough and professional suggestions. We have provided a detailed point-by-point response in the attached Word document for your review.

Reviewer 2 Report
Comments and Suggestions for Authors
First of all, I want to congratulate the authors for their efforts in this manuscript. The authors present the performance of different machine learning models to identify the adulteration of camel milk with cow milk. The topic falls within the journal's scope and is updated. In general terms, the paper has a good structure and solid results. There are some minor issues to be solved in order to ensure that the paper reaches the expected quality of the journal. Following, I include a series of comments aimed at enhancing the quality of the paper:
1. The abstract must be extended to reach 200-250 words. I suggest to include a short description of the problem in the first two sentences. Furthermore, the author can include more information about the material and methods and about the introduction of the problem.
2. Avoid using as a keyword, terms already used in the title. Deleted keywords included in the title and provided new keywords. Check all the provided keywords.
3. Add more keywords to have a minimum of 6 keywords.
4. In paragraph lines 60-70 the authors can include the use of machine learning to detect adulterated products based on other sources of data beyond the infrared spectrum. The following references are updated examples of the use of machine learning for detecting adulteration using different types of sensors.
a) (2023). Aromatic fingerprints: VOC analysis with E-nose and GC-MS for rapid detection of adulteration in sesame oil. Sensors, 23(14), 6294.
b) (2023). Proposal of a Gas Sensor-Based Device for Detecting Adulteration in Essential Oil of Cistus ladanifer. Sustainability, 15(4), 3357.
c) (2023). Application of residual neural networks to detect and quantify milk adulterations. Journal of Food Composition and Analysis, 105427.
5. In paragraph lines 71-79 it would be beneficial to include more information for the test bench, such as the used milt sources and the % of adulteration tested.
6. The sentence in lines 87-90 is too long. Please divide it into two sentences.
7. The sentence in lines 119-120 is missing the verb. Check the sentences.
8. In subsection 2.3, the authors have to clarify if the data has been normalized and mention the used equation.
9. When an acronym is defined, the first letter of each word in the acronym must be capitalized. Check the case of LV in which it is not done and similar cases such as CV.
10. Mention the number of folds selected for the cross-validation.
11. In subsection 2.4, I strongly recommend including the equations for bias, recovery, repeatability and precision since these terms can be confused with some commonly used metrics for machine learning analyses.
12. Is it possible to include in Table 1 the descriptive analyses of certain (or all) adulterated milk? Moreover, it would be beneficial to add the results of ANOVA to identify whether the differences between samples are statistically significant.
13. Check that some acronyms are repeated twice, such as CV for cross-validation and for coefficient of variation. Consider changing the acronym for any of them.
14. The sentence in lines 204-205 should include the citation of mentioned literature.
15. Concerning Figure 4 b, the authors should evaluate whether the points beyond the 5% can be considered outliers.
16. The authors have to compare the performance of their adulteration detection models with existing models in the literature in terms of metrics. The papers included in previous comments and the current literature in the paper can be used to generate a comparative table.
17. Future work should be added in an independent paragraph at the end of the conclusions.
Comments on the Quality of English LanguageCheck the previous comments
Author Response

(The authors gave the same response as above.)

Round 2
Reviewer 1 Report
Comments and Suggestions for Authors
Most of the remarks were taken in account.
I appreciate